# The Analysis of Relevant Gene Networks Based on Driver Genes in Breast Cancer

**DOI:** 10.3390/diagnostics12112882

**Published:** 2022-11-21

**Authors:** Luxuan Qu, Zhiqiong Wang, Hao Zhang, Zhongyang Wang, Caigang Liu, Wei Qian, Junchang Xin

**Affiliations:** 1School of Computer Science and Engineering, Northeastern University, Shenyang 110169, China; 2College of Medicine and Biological Information Engineering, Northeastern University, Shenyang 110169, China; 3Department of Breast Surgery, Cancer Hospital of China Medical University, Liaoning Cancer Hospital and Institute, Shenyang 110042, China; 4Department of Breast Surgery, Shengjing Hospital of China Medical University, Shenyang 110004, China; 5College of Engineering, The University of Texas at El Paso, El Paso, TX 79968, USA; 6Key Laboratory of Big Data Management and Analytics, Northeastern University, Shenyang 110169, China

**Keywords:** breast cancer, protein–protein interaction, mutual information, centrality analysis, survival analysis

## Abstract

Background: The occurrence and development of breast cancer has a strong correlation with a person’s genetics. Therefore, it is important to analyze the genetic factors of breast cancer for future development of potential targeted therapies from the genetic level. Methods: In this study, we complete an analysis of the relevant protein–protein interaction network relating to breast cancer. This includes three steps, which are breast cancer-relevant genes selection using mutual information method, protein–protein interaction network reconstruction based on the STRING database, and vital genes calculating by nodes centrality analysis. Results: The 230 breast cancer-relevant genes were chosen in gene selection to reconstruct the protein–protein interaction network and some vital genes were calculated by node centrality analyses. Node centrality analyses conducted with the top 10 and top 20 values of each metric found 19 and 39 statistically vital genes, respectively. In order to prove the biological significance of these vital genes, we carried out the survival analysis and DNA methylation analysis, inquired about the prognosis in other cancer tissues and the RNA expression level in breast cancer. The results all proved the validity of the selected genes. Conclusions: These genes could provide a valuable reference in clinical treatment among breast cancer patients.

## 1. Introduction

Genes are DNA or RNA fragments that carry genetic information and that can synthesize protein through a transcription–translation process or control the performance of individual organisms by affecting the synthesis and biology of human beings [1,2]. The incidence, development, prognosis, and drug resistance of common human diseases, such as malignant tumors and neurodegenerative diseases, can be attributed to the abnormal expression of genes [3,4]. For example, the activation of oncogenes could destroy the stability of the genome and cause cancer, or the inactivation of tumor suppressor genes could cause the genome to lose its role in inhibiting the growth of cancer cells [5]. Therefore, accurate identification and regulation of abnormal gene expression levels are one of the keys methods of treating diseases. In recent years, with the continuous emergence of clinical application of targeted therapies for different tumors, the treatment of malignant tumors had been greatly improved. Targeted therapeutic drugs play a very important role in precision medical treatment by regulating the expression levels of target genes for disease treatment [6,7].

Globally, breast cancer causes the highest number of malignant tumors in females, which seriously affects patients’ survival time and quality of life [8,9]. In the field of breast cancer treatment, Herceptin, a drug targeting the HER-2 protein, has achieved notable efficacy in prolonging the survival time of patients in patients with recurrent metastasis and receiving neoadjuvant therapy. However, the discovery of both driver genes and targeted genes often depends on the experimental experience of researchers and on the screening of genes and the running of biological experiments to verify the authenticity of their hypothesis [10]. This process leads to the low success rate of screening genes. In addition to being time-consuming, laborious, and resource-intensive, the biggest problem of this process is that it also leads to failure in the identification of key genes. The expressions of different genes are not isolated. One gene’s expression can influence the expression of other genes, whereas it is also influenced by other genes’ expressions in turn [11,12]. The interaction and mutual restriction constitute a protein–protein interaction network containing tens of thousands or even tens of millions of genes, and this complexity is far beyond brain’s reasoning ability [13]. By analyzing this network on a graph theory level, we can identify the key genes of gene–gene interactions, the network of upstream and downstream gene interactions, and the multi-gene common signaling pathways, providing valuable information on disease pathogenesis and treatment strategies.

In the area of breast cancer early detection, Computer-Aided Diagnosis [14,15], Machine Learning [16,17], and Deep Learning [18,19] have all achieved promising progress and results. However, the development of breast cancer is usually related to genetic factors, so it is necessary to conduct in-depth research on the genetic domain. The first step of breast cancer-related gene network analysis is gene selection, which selects the relevant genes relating to breast cancer. Presently, 90 driver genes in breast cancer have been identified [20,21]; this set of genes can be considered as the original gene set. After this set, some other relevant genes should also be selected, and the protein–protein interaction network should be established based on these two relevant genes sets. The present gene selection method is usually based on clustering or machine learning, which selects a certain category of data from the gene database. However, there is not a commonly accepted standard or evaluation criteria in the relevant gene selection process for clustering or machine learning based on the methods mentioned above. With this problem in mind, this paper proposed a breast cancer-relevant gene selection method based on mutual information. Then, we analyzed the protein–protein interaction network, counted the genes with high centrality in the network as a vital gene set, and explored and validated the functions of the genes in this gene set using bioinformatics analysis. The contributions of this paper can be summarized as follows:The mutual information method is used for the gene selection step, which selects breast cancer-relevant genes from the whole genome. Using this method, we selected 230 genes as the relevant genes for breast cancer.The protein–protein interaction network is built and analyzed based on the selected genes from the mutual information method. By analyzing the node centrality of the protein–protein interaction network, we obtained the important genes with important positions and connectivity in the network.Based on the vital genes, through survival analysis, DNA methylation analysis, and RNA expression level in breast cancer and the prognosis in other cancers, we found some genes that reduce the survival rate of breast cancer patients due to different expression levels and confirmed their biological significance.

## 2. Methods

### 2.1. The General Framework

The methods to accomplish relevant breast cancer genes network analysis should include three steps: gene selection, protein–protein interaction network modeling, and network analysis. We used the breast cancer gene expression data from the TCGA database, which includes 678 breast cancer patient samples and 23,760 genes. Firstly, we preprocessed the gene expression data, deleted the genes whose invalid values exceeded 50%, and replaced the missing values with the average value. Then, gene selection was performed based on the preprocessed gene expression data set. In the gene selection step, we chose the mutual information method and proposed a more simplified computation. For this step, we selected 230 genes, including the 90 genes which have been identified as a driver genes in breast cancer [20,21], as the relevant genes for breast cancer. Then, based on these 230 genes, the protein–protein interaction network was established based on the information obtained from the STRING database. After that, we analyzed the protein–protein interaction network with a centrality analysis of the nodes. The degree centrality (DC) [22], closeness centrality (CC) [23], betweenness centrality (BC) [24], and eigenvector centrality (EC) [25] were calculated. Based on these results, 19 and 39 genes that had the top 10 and top 20 values in the 4 metrics, respectively, were chosen as the vital genes. Finally, the survival analysis of these 19 genes and 39 genes showed that some differentially expressed genes affecting the prognosis of breast cancer. The process of the method is shown in Figure 1.

#### Mutual Information Method

The mutual information method is a useful information measurement method in information theory [26]. Mutual information is usually used to measure the reliability between two variables, *X* and *Y*; therefore, the correlation between two genes can be found in gene expression data. In gene expression data, a gene is represented by variable *X*, and the sample value of the same gene under different conditions can be represented as the value of variable *X*.

For a discrete variable *X*, the entropy H(X) is the average information amount from all messages received. It can be represented as:(1)$$H\left(X\right)=-\sum_{x\in X}p(x)logp(x)$$
where p(x) is the marginal possibility distribution function of vector *X*.

The joint entropy of *X* and *Y* can be represented as:(2)$$H\left(X,Y\right)=-\sum_{x\in X,y\in Y}p(x,y)logp(x,y)$$

MI can measure the reliability between two variables. Normally, the mutual information of two discrete random variables *X* and *Y* can be represented in the form of entropy:(3)I(X,Y)=H(X)+H(Y)−H(X,Y)

A higher value of mutual information indicates a closer correlation between two variables, while a lower value of mutual information indicates the anti-correlation between the two variables.

Here, the entropy is estimated with Gaussian kernel probability density estimator as follows [27]:(4)$$\begin{array}{c} P\left(X_{i}\right)=\frac{1}{N}\sum_{j=1}^{N}\frac{1}{{\left(2\pi \right)}^{n/2}{\left|C\right|}^{n/2}}\cdot \exp\left(-\frac{1}{2}{\left(X_{j}-X_{i}\right)}^{T}C^{-1}\left(X_{j}-X_{i}\right)\right) \end{array}$$
where *C* is the covariance matrix of variable *X*, $\left|C\right|$ is the determinant of matrix *C*, *N* is the number of samples, and *n* is the number of variables (genes) in *C*.

According to Equations (Equation 1) and (Equation 4), the entropy of variable *X* can be represented as follows:(5)$$H\left(X\right)=\log\left[{\left(2\pi e\right)}^{n/2}{\left|C\right|}^{1/2}\right]=\frac{1}{2}\log{\left(2\pi e\right)}^{n}\left|C\right|$$

According to Equation (Equation 5), Equation (Equation 3) can be transformed as:(6)$$I\left(X,Y\right)=\frac{1}{2}log\frac{\left|C(X)\right|\cdot \left|C(Y)\right|}{\left|C(X,Y)\right|}$$

Here, the computation of mutual information between two variables is simplified by the computation of the covariance, therefore resulting in an efficient formula for computing mutual information between two variables.

Compared to selecting genes, if we calculate each pair of genes in the whole genome based on the mutual information method, it will lead to a very large number of calculations. This would be a waste of time and meaningless, and more importantly, there is no selection standard. Therefore, we chose the 90 discovered driver genes in breast cancer [20,21] as the original gene sets, then selected a certain number of genes from the breast cancer gene expression data as the overall relevant genes for breast cancer. These data were then used to conduct protein–protein interaction network analysis. The steps of gene selection based on the mutual information method in Algorithm 1 are as follows.
**Algorithm 1** Gene selection based on Mutual Information.**Input:** Gene expression data $X\in {\left\{X_{i}\right\}}_{i=1}^{m}$; $Y\in {\left\{Y_{j}\right\}}_{j=1}^{n}$**Output:** MI values and the according Genes $\left(I_{j},Y_{j}\right)$**1** **for**j=1 to *n*
**do****2**  **for**i=1 to *m*
**do****3**    calculate $\left|C(X_{i})\right|$, $\left|C(Y_{j})\right|$, $\left|C(X_{i},Y_{j})\right|$;**4**    calculate $I(X_{i},Y_{j})$ using Equation (Equation 6);**5**    **if**
$I(X_{i},Y_{j})$ is the max **do****6**     save $I(X_{i},Y_{j})$ to $I_{j}$;**7**   
**end if****8**   save several $\left(I_{j},Y_{j}\right)$ according to the maximum values of $I_{j}$;**9** 
**return**

The gene expression data are divided into two matrices, X and Y. Matrix X contains the gene expression data of the 90 driver genes of breast cancer, while matrix Y contains the rest of the gene expression data. The outcome is listed as a number of records from matrix Y, which has the highest correlation values (this indicates that they have the highest mutual information value) with any of the expressions from matrix X. Then, we read one line of data from matrix Y and compute the mutual information value with each line recorded in matrix X (lines 1–4). Afterwards, we save the data with the highest mutual information value in $I_{j}$ (lines 5–7). After completing all mutual information computation in matrix Y, we send the highest $I_{j}$ records and their corresponding $Y_{j}$ to $\left(I_{j},Y_{j}\right)$ and display the final results (lines 8–9).

### 2.2. Node Centrality

In the network analysis, centrality is an index to judge and quantify the importance of nodes [28]. Thus, we chose four aspects, namely, degree centrality (DC) [22], closeness centrality (CC) [23], betweenness centrality (BC) [24], and eigenvector centrality (EC) [25], to evaluate the nodes in our gene regulatory networks. The nodes in the networks represent the 340 selected genes.

DC is a simple measurement that counts how many neighbors a node has and describes the direct influence of the nodes in the networks. It can be defined as follows:(7)$$DC_{i}=\frac{k_{i}}{N-1}$$

Closeness is based on the length of the average shortest path between a vertex and all vertices in the networks. CC describes how easy it is for a node to reach other nodes in the networks. The CC of nodes can be represented by the equation below:(8)$$CC_{i}=\frac{N}{\sum_{j=1}^{N}d_{ij}}$$
where $d_{ij}$ is the distance between node *i* and node *j*.

BC counts the fraction of shortest paths going through a given node. It describes the control ability of a node through which node pairs transmit information along the shortest path in the networks. More precisely, the BC of a node *i* can be described as follows:(9)$$BC_{i}=\sum_{s\neq i\neq t}\frac{n_{st}^{i}}{g_{st}}$$
where $n_{st}^{i}$ is the number of shortest paths from node *s* to node *t* going through node *i*, and $g_{st}$ is the total number of shortest paths from *s* to *t*.

The importance of a node depends on both the number and importance of its neighbors. EC is used to describe this property:(10)$$x_{i}=\frac{1}{\lambda }\sum_{i=1}^{N}a_{ij}x_{j}$$
where $a_{ij}=1$ if vertices *i* and *j* are connected by an edge and $a_{ij}=0$ if they are not; λ is the largest eigenvalue of $\sum a_{ij}$.

## 3. Results

In the gene selection step, the mutual information values calculated by the other remaining genes and driver genes are shown in Figure 2a. The scatter points in the figure are the calculated mutual information values of each genes, and the red lines are the fitting curves of these values. The derivative of the fitting function is obtained according to the trend of the calculated results, and α=−0.0005 is taken as the threshold value in the obtained results in Figure 2b. Through the set threshold, 140 genes with high mutual information value, that is, strong correlation with a driver gene, were selected.

Based on the selected breast cancer-related gene set in the gene selection step, the interactions of 230 genes were queried in the STRING database, and the protein–protein interaction network of these genes was reconstructed. The reconstruction results are shown in Figure 3.

The corresponding degree centrality, closeness centrality, betweenness centrality, and eigenvector centrality values of the 230 genes are shown in Figure 4. For convenience, the genes were numbered from 1 to 230. There are several genes with a high value, and the remaining ones are generally lower in Figure 4. Thus, the higher-value genes with higher metrics are more valuable and important to analyze in the protein–protein interaction network.

Based on the results of Figure 4, we summarized the top 10 genes with the highest value among the 230 genes. First, the genes were sorted by the value from largest to smallest. Then, the top 10 genes with the largest values were selected and shown in Table 1. There are 19 genes in Table 1, including 10 driver genes and 9 selected genes.

In addition, a total of 39 of the top 20 genes with the highest value among the 230 genes are also summarized and shown in Table 2, including 19 driver genes and 20 selected genes. These calculated 19 and 39 genes are called the top 10 and top 20 vital genes.

The 3650-day (10-year) survival analysis statistics of these 19 vital genes of the top 10 were carried out, and five genes have a log rank p<0.05, which are CDK1, CCNB1, BUB1, BUB1B, and KIF20A. The survival curves of these genes are shown in Figure 5.

The 3650-day (10-year) survival analysis statistics of the 39 vital genes of the top 20 were carried out, and 12 genes with a log rank p<0.05 were obtained, including CDK1, CCNB1, BUB1, BUB1B, KIF23, CCNB2, KIF20A, KIF4A, MELK, RAD51, HRAS, and CEP55. Because the genes in top 20 genes must contain that the top 10 genes, so that the 5 genes with significant differences in survival rate among the top 10 genes must also be included in the 12 genes in this experiment. Therefore, we only list the survival curves of the remaining 7 genes in the top 20, and the survival curves of these 7 genes are shown in Figure 6.

The DNA methylation analysis [29,30] also concluded that 11 of the top 10 genes in Table 1, which are CCNA2, CCNB1, TP53, BRCA1, TOP2A, CCND1, AKT1, CREBBP, SMAD4, ESR1, and CENPE, had a higher level of DNA methylation in breast cancer. The detailed CpGs in Figure 7 show that the following genese displayed higher expression levels in breast cancer: cg07263562 of CCNA2; cg13849825, cg13647309, cg17668562 of CCNB1; cg10792831, cg16397722 of TP53; cg07054526, cg16029534 of BRCA1; cg22935319 of TOP2A; cg11234767, cg15974867 of CCND1; cg02072813, cg06934468, cg10100767, cg01694276, cg20923444 of AKT1; cg16560077, cg01963870, cg27390443, cg27318635, cg03140190, cg05898629 of CREBBP; cg00400189 of SMAD4; cg12209876, cg03732055, cg09414638 of ESR1; cg21346648, cg24651824, cg27443373 of CENPE.

Based on the 39 genes which are the vital genes of the top 20 in Table 2, we searched the RNA expression of these genes in breast cancer on the proteinatlas database [31,32,33]. All 39 genes were expressed in breast cancer (FPKM > 1). In addition, we also searched for prognostic markers of these genes in other cancer tissues and found that 34 genes had significant functions (*p* < 0.001). The results are shown in Table 3, where (−) indicates that the gene has an unfavorable prognostic marker in the analyzed cancer, and (+) indicates that the gene is a favorable prognostic marker in the analyzed cancer.

## 4. Discussion

In this study, we have selected 90 genes that have previously been proven to play important roles in the biological behavior of breast cancer [20,21]. The functions of these genes have been verified by molecular biology, laboratory animal science, and other methods, which are major breakthroughs in breast cancer occurrence investigations, as well as in investigations into the development, prognosis, and drug resistance of breast cancer. By identifying the interaction network of abnormal genes, it is possible to understand all kinds of cell signals from extracellular signals to nucleus ones. This leads us to trying to understand the whole process of changes in biological behavior, so we can better identify the key regulatory nodes of the transduction pathway and so we can point out potential candidate target genes for the development of new targeted drugs. We seek the genes that are most closely associated with the expression of these 90 genes in the whole genome and determine the scope of the problem in 140 genes by defining a threshold and filtering out the rest. Thus, we can determine the interaction environment of these 90 important genes and other vital genes in the protein–protein interaction network. We limited the scope of gene fishing in order to focus on these 90 genes and their interactive genes and to reduce difficulties in performing the functional analysis and verification of subsequent genes.

By querying the interactions of these selected genes in the STRING database, a protein–protein interaction network was obtained. From this protein–protein interaction network, we can see that the interaction between genes is complex: some genes are closely related to other genes, while the relationship between other genes is sparse. This is very similar to the interaction relationship of the whole genome network, so we can assume that the selected genes are a summary network diagram centered on driver genes, that is to say, the interaction relationship between genes may be indirectly connected. The direct or indirect relationships between these selected genes are also very likely to exist in the upstream and downstream of driver genes or metabolic pathways. One of the purposes of selecting these genes is to construct a metabolic pathway from target genes to driver genes or pathogenic genes and then provide some valuable gene sets or target genes for drug research and development or clinical treatment.

After completing the previous steps, we conducted sorting assignments for the 230 genes based on the four metrics, which are degree centrality, closeness centrality, betweenness centrality, and eigenvector centrality, and the top 10 and top 20 genes in each metric were selected. The main purpose of this task is to find the centralized nodes, the most important pathways, and the closest networks among these genes. Finally, 19 genes and 39 genes were counted in the top 10 and 20 genes, respectively, of the four metrics. The vital genes of the top 20 genes selected by node centrality analysis were verified on the proteinatlas database using multiple verification methods, which confirmed that all the selected genes had RNA expression in breast cancer. Moreover, by further exploring the prognostic marker information of these genes in other cancer tissues, we can see that most of these genes have poor prognosis in some other genes. In other words, the abnormal expression of these genes may lead to the recurrence and metastasis of diseases. At the same time, some genes also show good prognosis in several cancers, which also provides more information for clinical treatment and research of cancer diseases. Therefore, most of these vital genes obtained by node centrality analysis are genes with significant prognostic functions. This shows the effectiveness of the network analysis method we chose.

Finally, based on the statistics of 19 and 39 vital genes from node centrality analysis, the survival analysis experiments were carried out. Among them, 5 of 19 genes have significant expression levels with log rank *p* values <0.05, indicating that these 5 genes will have an impact on the prognosis of breast cancer when they are highly expressed. In addition, 11 of the 39 genes will reduce the survival rate when they are highly expressed, while 1 gene will affect the prognosis of breast cancer when it is under-expressed. Among these genes are BUB1B and HRAS, which have previously been proven to play important roles in the biological behavior of breast cancer, and the other genes are all genes we fished out. This shows that our gene fishing and network analysis method not only focuses on biologically significant genes, but also selects other genes that are closely related to them and which will have an important impact on the occurrence, development, and treatment of breast cancer. It shows that our method of gene fishing is effective and efficient, and our method of network analysis can find some gene nodes that play an important role in disease treatment in the network and cannot be observed by biological experiments or human eyes.

In the future, we should pay more attention to the signal transduction pathways of genes, try to simulate the upstream and downstream and metabolic pathways of gene regulatory networks, and screen out gene sets that are more in line with direct regulation and indirect regulation by decaying the degree of association layer by layer. By doing so, we can include more related genes and incorporate them into the networks, which would help us to observe the more comprehensive networks and analyze some bypasses of important signal transduction pathways. Because of the high conservatism of signal transduction pathways in human tissue cells, there are similar or highly similar signal transduction networks in the diseases of different systems. Due to the current difficulty of reaching the desired level of tissue and organ specificity in the application of drugs, there are often some adverse multi-system reactions with various degrees in disease treatment. Based on the multi-gene common signaling pathway, we can analyze the main path and the bypass state of the signal transduction pathway after drug action, and analyze the changes and degrees of related gene expression to predict the types, possibilities, and severity of adverse drug reactions in drug development, clinical trials, and clinical applications. This could become a reference for measuring the benefits and risks of drugs. Meanwhile, bypassing the signal pathway could be truncated to avoid the adverse effects caused by the bypass. It would be a basis for the prevention and effective control of adverse drug reactions and would be helpful in clinical decision making in the future.

## 5. Conclusions

Like many other cancers, the development of breast cancer is also related to genetic factors. By looking into the relationship of genetics with the development cancer, it is possible to regulate and control the expression of oncogenes, which may have profound influence in cancer prevention and treatment. This is the most economical and practical way for both patients and doctors to make full use of the existing drugs targeting the regulation of gene expression. We have selected some of the relevant genes from breast cancer and conducted protein–protein interaction network analyses. First, we used the mutual information method to select the relevant genes in addition to the driver genes in breast cancer, which were determined previously. Second, by modeling the protein–protein interaction network using the STRING database, we were able to obtain a clear picture of the relationship between genes. Finally, for our analysis, we chose genes with higher values of node centrality as the vital genes, then conducted survival analysis and DNA methylation analysis in breast cancer and prognosis analysis in other cancers using these vital genes. Through analyzing the protein–protein interaction network, we found some genes related to poor prognosis and higher methylation due to different expressions in breast cancer.

## Figures and Tables

**Figure 1 diagnostics-12-02882-f001:**
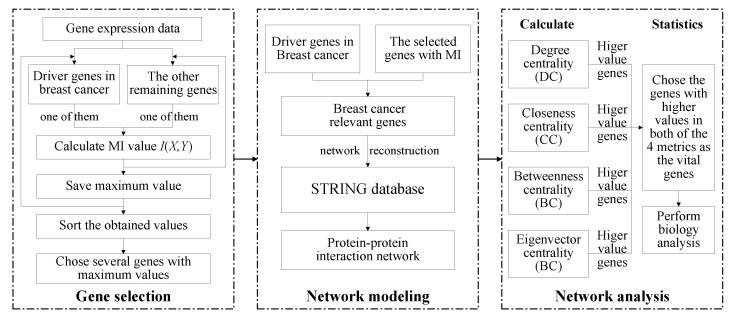
The process of the method. The three steps of the process are breast cancer gene selection, protein–protein interaction network modeling, and network analysis.

**Figure 2 diagnostics-12-02882-f002:**
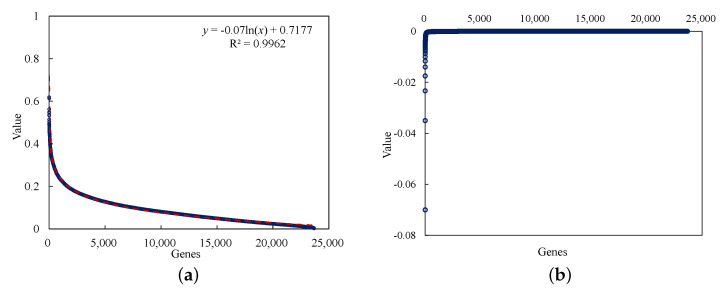
The results of gene selection step. (**a**) Mutual information values. The calculated values of all other remaining genes and driver genes, which, in function *y*, is the fitting curve equation, and R${}^{2}$ is the coefficient of determination, meaning the higher the fitting degree, the closer to 1. (**b**) Threshold. All values are derived based on the obtained fitting curve function *y* in (**a**) and sorted according to the resulting values.

**Figure 3 diagnostics-12-02882-f003:**
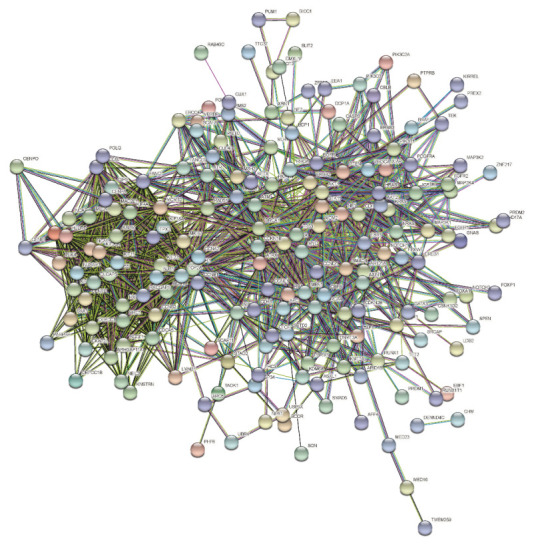
The protein–protein interaction network of breast cancer-relevant genes. Node represents gene, and edge represents the interaction between two genes.

**Figure 4 diagnostics-12-02882-f004:**
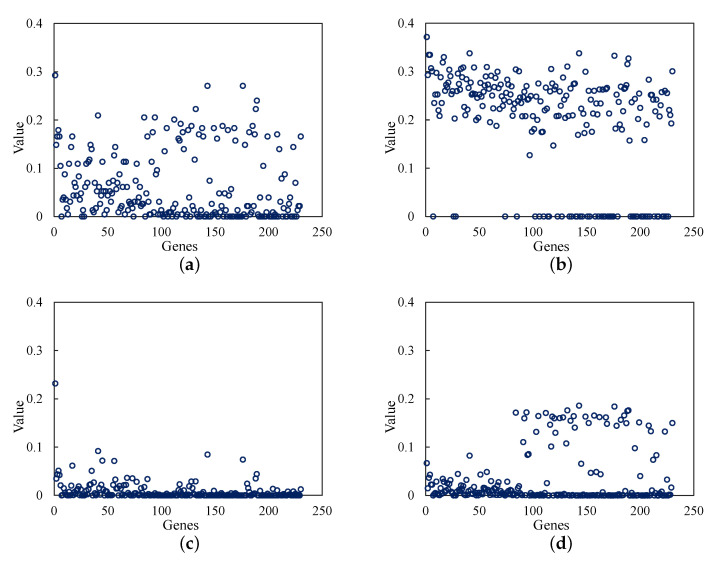
Node centrality analysis of 230 genes. The abscissa is a gene, which is represented by a number, and the ordinate is the result value of a gene calculated by the corresponding node centrality metric. (**a**) Degree centrality; (**b**) Closeness centrality; (**c**) Betweenness centrality; (**d**) Eigenvector centrality.

**Figure 5 diagnostics-12-02882-f005:**
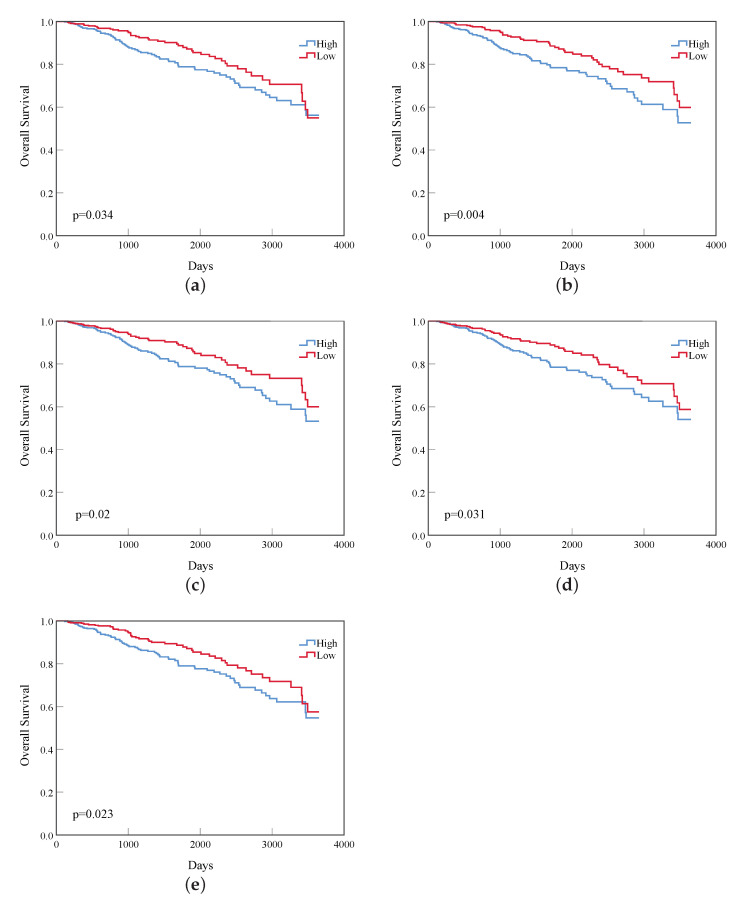
The gene survival curves of the top 10 genes. The blue line indicates that the expression value is higher than the median, and the red line indicates that the expression value is lower than the median. P-value is the result of log-rank test (*p* < 0.05 means the result has the significant). (**a**) CDK1; (**b**) CCNB1; (**c**) BUB1; (**d**) BUB1B; (**e**) KIF20A.

**Figure 6 diagnostics-12-02882-f006:**
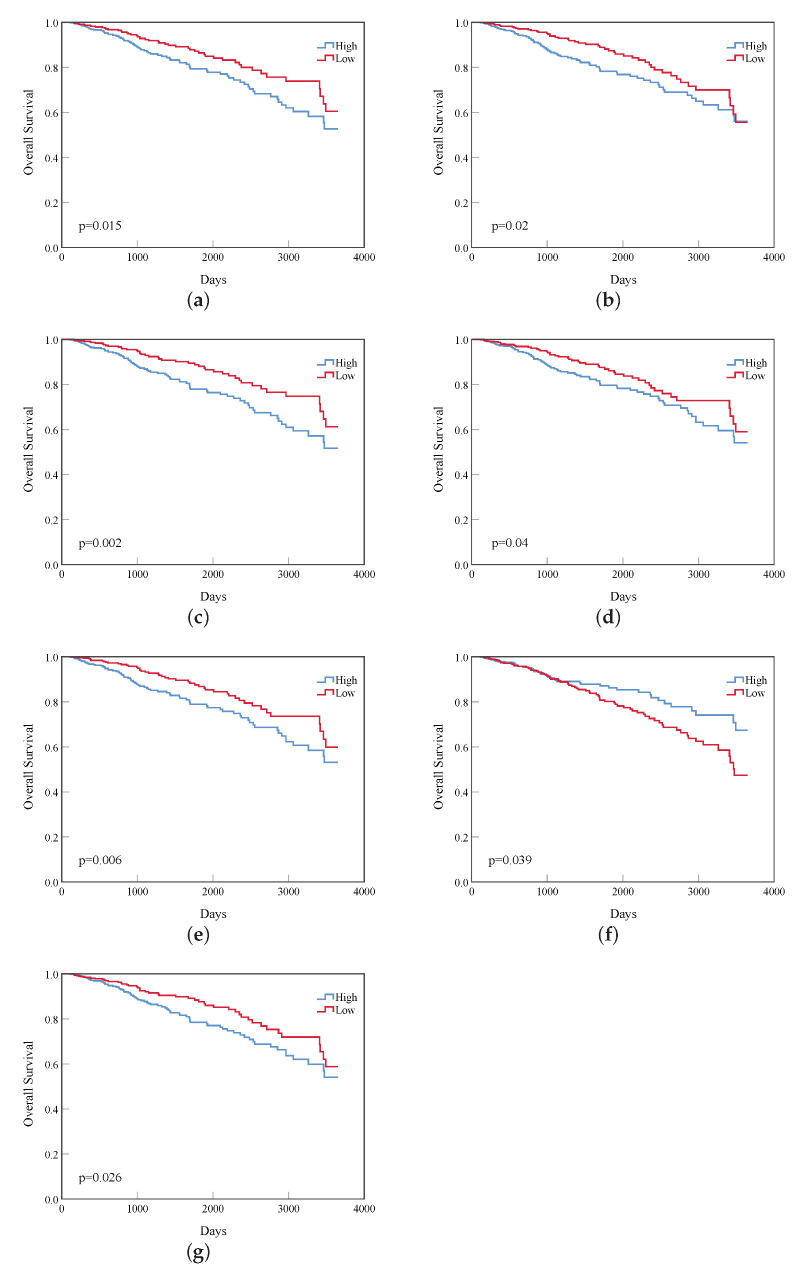
The remaining 7 genes’ survival curves of the top 20 genes. (**a**) KIF23; (**b**) CCNB2; (**c**) KIF4A; (**d**) MELK; (**e**) RAD51; (**f**) HRAS; (**g**) CEP55.

**Figure 7 diagnostics-12-02882-f007:**
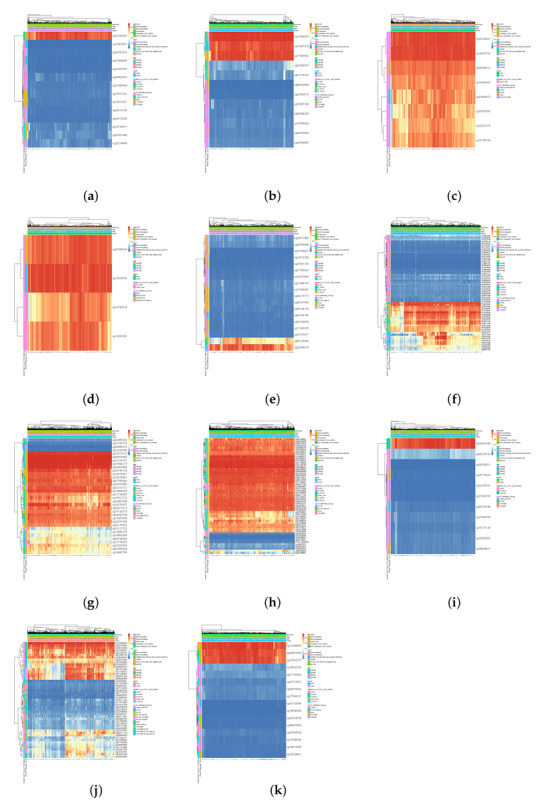
Heatmap of DNA methylation expression levels of top 10 genes in breast cancer using the MethSurv platform. Methylation levels (1 = fully methylated; 0 = fully unmethylated) are shown as a continuous variable from a blue to red color. Rows correspond to the CpGs, and the columns correspond to the patients. (**a**) CCNA2; (**b**) CCNB1; (**c**) TP53; (**d**) BRCA1; (**e**) TOP2A; (**f**) CCND1; (**g**) AKT1; (**h**) CREBBP; (**i**) SMAD4; (**j**) ESR1; (**k**) CENPE.

**Table 1 diagnostics-12-02882-t001:** Top 10 genes in four metrics of node centrality.

Metrics	Top 10 Genes
Degree Centrality	TP53, CCNA2, CDK1, CCNB1, BUB1, TOP2A, BRCA1, BUB1B, KIF11, NCAPG
Closeness Centrality	TP53, BRCA1, CCNA2, MYC, CCND1, CDK1, AKT1, CCNB1, CDKN2A, TOP2A
Betweenness Centrality	TP53, BRCA1, CCNA2, CDK1, CREBBP, SMAD4, AKT1, CCND1, ESR1, CCNB1
Eigenvector Centrality	CCNA2, CDK1, BUB1, CCNB1, TOP2A, KIF11, BUB1B, NCAPG, KIF20A, CENPE

**Table 2 diagnostics-12-02882-t002:** Top 20 genes in four metrics of node centrality.

Metrics	Top 20 Genes
Degree Centrality	TP53, CCNA2 CDK1, CCNB1, BUB1, TOP2A, BRCA1, BUB1B, KIF11, NCAPG, CCNB2, KIF23, CENPE, KIF20A, KIF4A, ASPM, TPX2, DLGAP5, CCND1, KIF15
Closeness Centrality	TP53, BRCA1, CCNA2, MYC, CCND1, CDK1, AKT1, CCNB1, CDKN2A, TOP2A, BUB1, ATM, ESR1, CREBBP, PTEN, EGFR, RAD51, BUB1B, MDM2, ERBB2
Betweenness Centrality	TP53, BRCA1, CCNA2, CDK1, REBBP, SMAD4, AKT1, CCND1, ESR1, CCNB1, MYC, PTEN, STAG2, CNOT3, TOP2A, PIK3CA, HRAS, ATM, BUB1, KIF23
Eigenvector Centrality	CCNA2, CDK1, BUB1, CCNB1, TOP2A, KIF11, BUB1B, NCAPG, KIF20A, CENPE, KIF4A, ASPM, CCNB2, TPX2, MELK, DLGAP5, KIF23, KIF15, CEP55, NUSAP1

**Table 3 diagnostics-12-02882-t003:** Prognostic marker information in cancer tissue of the top 20 genes.

Gene	Prognostic Marker in Cancer
CCNA2	renal cancer(−); pancreatic cancer(−); liver cancer(−); lung cancer(−); endometrial cancer(−)
CDK1	renal cancer(−); liver cancer(−); pancreatic cancer(−); lung cancer(−); cervical cancer(+)
CCNB1	renal cancer(−); liver cancer(−); lung cancer(−)
TOP2A	renal cancer(−); liver cancer(−); pancreatic cancer(−); lung cancer(−)
BUB1	liver cancer(−); pancreatic cancer(−); endometrial cancer(−); lung cancer(−)
TP53	endometrial cancer(+); prostate cancer(−)
BUB1B	liver cancer(-); pancreatic cancer(−); lung cancer(−)
CCND1	pancreatic cancer(−); head and neck cancer(−)
KIF23	liver cancer(−); pancreatic cancer(−); endometrial cancer(−)
MYC	renal cancer(−); urothelial cancer(−); ovarian cancer(−)
KIF11	renal cancer(−); liver cancer(−); pancreatic cancer(−); lung cancer(−)
NCAPG	liver cancer(−); pancreatic cancer(−); endometrial cancer(−)
CCNB2	renal cancer(−); pancreatic cancer(−); melanoma(-); liver cancer(−); lung cancer(−)
KIF20A	renal cancer(−); liver cancer(−); pancreatic cancer(−); lung cancer(−)
KIF4A	liver cancer(−); pancreatic cancer(−);
ASPM	liver cancer(−); endometrial cancer(−); pancreatic cancer(−); lung cancer(−)
TPX2	renal cancer(−); liver cancer(−); endometrial cancer(−); pancreatic cancer(−); lung cancer(−)
DLGAP5	liver cancer(−); pancreatic cancer (−); endometrial cancer(−); lung cancer(−)
KIF15	colorectal cancer(+)
ESR1	endometrial cancer(+)
CREBBP	renal cancer(+)
PTEN	renal cancer(−)
AKT1	renal cancer(+); ovarian cancer(+)
SMAD4	renal cancer(+)
CDKN2A	endometrial cancer(−); renal cancer(−); liver cancer(−); head and neck cancer(+)
CNOT3	liver cancer(−); renal cancer(−)
MELK	renal cancer(−); liver cancer(−); lung cancer(−); pancreatic cancer(−)
EGFR	urothelial cancer(−)
RAD51	breast cancer(−); liver cancer(−)
HRAS	liver cancer(−)
MDM2	endometrial cancer(+); cervical cancer(+)
CEP55	renal cancer(−); liver cancer(−); pancreatic cancer(−); lung cancer(−); stomach cancer(+)
ERBB2	renal cancer(+); endometrial cancer(−); pancreatic cancer(−);
NUSAP1	renal cancer(−); pancreatic cancer(−)

## Data Availability

The datasets generated during the current study are available in the [TCGA] repository.

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
