# Peer review of "The Analysis of Relevant Gene Networks Based on Driver Genes in Breast Cancer"

_diagnostics, 2022, doi:10.3390/diagnostics12112882_

Round 1

Reviewer 1 Report

In the present study, the authors conducted multistep analysis of breast cancer relevant gene network by mutual information method.  In addition, the authors obtained the top 10 and top 20 genes with connectivity in the network. Through analyzing the network, the authors found some genes related to poor prognosis of breast cancer.

Although the overall structure of the manuscript seems to be well thought-off, and methods are justified, some changes need to be made before it is published.

Minor points

1.      The introduction is very detailed and thorough, including a description of the manuscript structure of the (see lines 81-86) which is unnecessary.

2.     In the discussion, the authors paid more attention to the gene network technology, its advantages and implications for clinical practice. The authors can revise this section and focused more on selected gene interaction and their signal transduction pathway networks.

3.     The authors should carefully check all text and remove any repeats (for example line 92).

Reviewer 2 Report

The author revealed selected some of the relevant genes from breast cancer and conducted the protein-protein interaction network analysis. Finally, they chose genes with higher values of node centrality as the vital genes, then conduct survival analysis using these vital genes. Through analyzing the network, they found some genes related to poor prognosis due to different expressions in breast cancer. Albeit, I consider these findings would guide future clinical interventions to a certain extent, I still have some suggestions.

1, All figures are highly professional, and the authors should guide the readers to the meaning of the images appropriately; otherwise, it is likely to cause misunderstandings. Therefore, I suggest that the author consider revising these figure legends again.

2, In Table 2. The author showed 20 genes in four metrics of node centrality. However, it is worth exploring and validating their data via proteinatlas (https://www.proteinatlas.org) or  kmplot (https://kmplot.com) database, and discussing these data as well as these methodologies for cancer recurrence or metastasis in the manuscript (PMID: 25613900, 32064155, 5626291)

3, The authors gave a general answer on gene expression, is there any evidence of different roles in cancer phenotypes of these genes either in Table 1 or Table 2? Please perform pertinent bioinformatic analyses and provide examples of studies investigating miRNA alteration or DNA methylation (https://biit.cs.ut.ee/methsurv/) (PMID: 29264942, 34834441). 

4, There are few typo issues for the authors to pay attention to, please also unify the writing of scientific terms. “Italic, capital”?  The font is too small for some of the current figures, please also revise these figures, meanwhile, the manuscript also needs English proofreading.
